

# Effects of cleaning sports mouthguards with ethylene-vinyl acetate on oral bacteria

Hiroki Hayashi[1,2], Yoshikazu Naiki[1], Masahiro Murakami[1,3], Akihiro Oishi[1], Rihoko Takeuchi[1,3], Masayoshi Nakagawa[2], Suguru Kimoto[2,3], Yoshiaki Hasegawa[1] and Akizumi Araki[2]

[1] Department of Microbiology, School of Dentistry, Aichi Gakuin University, Nagoya, Japan
[2] Department of Fixed Prosthodontics and Oral Implantology, School of Dentistry, Aichi Gakuin University, Nagoya, Japan
[3] Department of Gerodontology and Home Care Dentistry, School of Dentistry, Aichi Gakuin University, Nagoya, Japan

Corresponding author
Yoshiaki Hasegawa,
yhase@dpc.agu.ac.jp

## ABSTRACT

**Background.** Sports mouthguards, worn in the oral cavity to prevent sports injuries, are constantly exposed to various microorganisms that cause oral infections. Hence, the optimal cleaning methods for sports mouthguards have been thoroughly examined. In this study, we evaluated the efficiency of cleaning effects with a mouthguard cleaner (MC) on microbial biofilm formation in sports mouthguards *in vitro* and *in vivo*.

**Methods.** We evaluated the cleaning effects of the discs produced by ethylene-vinyl acetate (EVA) on bacterial biofilms formed by the commensal bacterium *Streptococcus oralis,* the cariogenic bacterium *Streptococcus mutans*, and the opportunistic pathogen *Staphylococcus aureus in vitro*. EVA discs with biofilm were subjected to sterile distilled water (CTRL) and ultrasonic washing (UW), followed by treatment with MC and sodium hypochlorite (NaClO) as positive controls. Thereafter, the viable bacterial cell counts were determined. The bacteria adhering to the sheets before and after the treatment were observed under an electron microscope. The degree of cleanliness and measurement of viable microbial cell counts for total bacteria, *Streptococci* and *Candida*, opportunistic fungi, were evaluated on the used experimental sports mouthguards with and without UW and MC treatment *in vivo*.

**Results.** The number of bacterial cells significantly decreased against all the tested biofilm bacteria upon treatment with MC, compared with CTRL and UW. Electron microscopy analysis revealed the biofilm formation by all bacteria on the EVA discs before cleaning. We observed fewer bacteria on the EVA discs treated with MC than those treated with CTRL and UW. Furthermore, the degree of cleanliness of the used experimental sports mouthguards cleaned using MC was significantly higher than that of the CTRL-treated mouthguards. Moreover, the viable microbial cell counts on the used experimental sports mouthguard were considerably lower than those on the CTRL ones.

**Conclusion.** The cleaning effect of MC against oral bacteria was more effective than that of UW. MC treatment might have a potential future application as a cleaning method for sports mouthguards to protect athletes from oral infection.

# INTRODUCTION

Sports mouthguards help prevent not only traumatic injuries, such as broken teeth (*Bergman et al., 2017*; *Cicek et al., 2021*) but also concussions when worn in the mouth (*Nakajima et al., 2017*; *Ono et al., 2020*). The number of athletes using sports mouthguards is increasing yearly, as many sports organizations have made the use of sports mouthguards mandatory as a safety measure. Ethylene-vinyl acetate (EVA) is the primary material used in the production of sports mouthguards.

Oral cavity diseases, such as dental caries and periodontitis, are caused by bacterial infections that colonize the oral tissues through biofilm formation. Biofilm-forming bacteria tightly bind to solid material surfaces in organisms through a net of extracellular matrices consisting of peptides and sugars (*Høiby, 2014*). The cariogenic bacterium *Streptococcus mutans* (*S. mutans*) and the oral commensal bacterium *Streptococcus oralis* (*S. oralis*), a gram-positive facultatively anaerobic coccus, firmly form biofilms on teeth (*Nobbs, Lamont & Jenkinson, 2009*). *Staphylococcus aureus* (*S. aureus*), a gram-positive facultative anaerobic coccus, is an opportunistic pathogen (*Sedarat & Taylor-Robinson, 2022*) and its multidrug-resistant strains, such as methicillin-resistant *S. aureus*, lead to critical medical problems worldwide owing to the lack of effective drugs against them (*Shrivastava, Shrivastava & Ramasamy, 2018*). Although *S. aureus* is not always detected in the oral cavity, it causes refractory inflammation in the oral mucosa and root canal because it can survive disinfection and antibiotic treatment in dental therapy procedures (*Smith, Jackson & Bagg, 2001*).

Sports mouthguards are worn in the mouth for extended periods, resulting in contamination with oral bacteria. Unsanitary sports mouthguards contaminated with oral bacteria can cause malodor and lead to periodontal disease and dental caries. *S. oralis* was also reported to persist for up to 14 days after wearing a sports mouthguard for only 5 min (*Ogawa et al., 2012*). Thus, daily cleaning of sports mouthguards, maintaining their cleanliness, and establishing appropriate cleaning methods are warranted to eliminate bacterial contamination.

There are two types of cleaning methods for sports mouthguards, mechanical and chemical (*Glass et al., 2011*; *Hasegawa, Sato & Kimoto, 2020*; *Tanabe et al., 2021*). Mechanical cleaning methods include the use of a soft brush, which might cause microscopic scratches on the surface of an EVA sheet, leading to bacterial contamination. Furthermore, similar to dentures, some areas of sports mouthguards are difficult to clean, and physical cleaning methods are limited. In contrast, *Glass et al. (2011)* reported that daily cleaning with cleaning agents could reduce microbial contamination on sports mouthguards. *Tanabe et al. (2021)* also reported that more than 99% of *Streptococus sobrinus* cultured on EVA were killed 60 s after spraying with a spray-type cleaning agent. However, no cleaning method with standardized guidelines has been developed thus far in Japan.

The development of standardized guidelines for mouthguard cleaning requires original research focusing on several perspectives such as bactericidal effects. Thus, we investigated the effect of a commercial cleaner (Aligner Retainer Cleaner, Morimura, Japan) on EVA sports mouth guards using *in vitro* and *in vivo* studies to help develop appropriate cleaning method guidelines.

*In vitro*, biofilms of various oral microorganisms were formed on discs prepared according to a standard method, and the cleaning efficacy of the samples was evaluated by measuring the colony forming units (CFU). After cleaning, the EVA surface was observed under an electron microscope to evaluate the efficiency of the cleaner. The degree of cleanliness and measurement of CFU on the used experimental sports mouthguard was evaluated *in vivo*.

## MATERIALS & METHODS

### *In vitro* study
#### *Preparation of EVA specimen sheets*
The impression of a plastic disc with a diameter of 100 mm was taken with alginate impression material, and a stone model was fabricated. The EVA sheet was heated and pressed over the stone model using a vacuum pressure machine. The disc sheet was removed from the stone model and cut using a metal hole punch with a diameter of five mm. A vinyl acetate disc of five mm diameter was used as the specimen. The specimens were sterilized using ethylene oxide.

#### *Bacterial strains and culture conditions*
The strains used in this study were *S. mutans* NCTC 10449, *S. oralis* ATCC 9811, and *S. aureus* FDA209P. All bacterial strains were inoculated on BHI agar medium and incubated untill they reached the quiescent stage on the growth curve at 37 °C under anaerobic conditions (80% $N_2$, 10% $H_2$, 10% $CO_2$). A sucrose-containing medium was used to induce biofilm formation. It is well known that *S. mutans* forms a strong biofilm by secreting water-insoluble glucan when grown in a sucrose-containing medium (*Hamada & Slade, 1980*), and we reported that this phenomenon also occurred with *S. aureus* FDA 209P (*Murakami et al., 2021*).

#### *Examination of sterilization effect on EVA discs for biofilm bacteria*
For *S. mutans,* twenty-four EVA discs were immersed in BHI broth containing 5% sucrose, inoculated in the medium, and incubated for 24 h to form biofilms. Twenty-four EVA discs with biofilm formation were transferred evenly into four 250-ml, 11-cm high beakers. The six EVA discs in each beaker were treated with 150 ml of sterile distilled water at 40 °C (CTRL), ultrasonic washing (UW), tablet mouthguard cleaner (MC; Aligner Retainer Cleaner, Morimura, Japan), or 250 ppm sodium hypochlorite (NaClO). For *S oralis* and *S. aureus*, bacterial biofilms were formed by the same procedure as for *S. mutans*. The control group was immersed in sterile distilled water for 10 min. In the UW group, the EVA discs were immersed in sterile distilled water and treated in an ultrasonic cleaner (ULTRASONIC CLEANER UC-0205A; NIPPON Kagaku KIKAI, Tokyo, Japan) at 48 W

and 27 kHz for 10 min. In the MC group, EVA discs were immersed in sterile distilled water at 40 °C with detergent and disinfected for 10 min. In the NaClO group, which was used as a positive control, EVA discs were immersed in 150 ml of 250 ppm NaClO at 40 °C and disinfected for 10 min. The water temperature for all groups was unified at 40 °C to follow the instructions for the tablet mouthguard detergent. The mouthguard cleaner used in this study is a commercially available cleaning agent whose main ingredient is cetylpyridinium chloride, a widely used cleaning agent that has been reported to have few harmful effects on living organisms.

After rinsing with sterile distilled water, the discs were vortexed in 15-ml test tubes containing 1 ml of PBS for 20 s. The discs in the test tubes were treated with an ultrasonic cleaner for 10 min to release the bacteria attached to the discs into the test tubes. The bacterial suspension obtained was inoculated into BHI agar medium and incubated for 24 h to form colonies. Finally, the number of bacterial colonies was counted. Serial dilutions of the bacterial suspensions were spread on agar plates, and CFUs were enumerated after incubation. Each of the experiments, with three parallels, were performed in triplicate.

### Observation by a scanning electron microscope (SEM)
After rinsing one disc from each of the four groups with sterile distilled water, the bacteria were fixed by immersion in a phosphate buffer solution of 4% paraformaldehyde and 2% glutaraldehyde for 24 h. The discs were then dehydrated sequentially in ethanol concentrations of 20%, 30%, 40%, 50%, 60%, 70%, 80%, 90%, and 100%. The bacteria on the discs were observed using a scanning electron microscope (JXA-iHP200F; Nihondenshi, Tokyo, Japan) at 5,000 × magnification.

### In vivo study
#### Participants
Ten healthy volunteers were recruited for this study, whose ages ranged from 27 to 33 ($28.8 \pm 2.4$) years. Four patients were male and six were female. The study protocol was reviewed and approved by an Ethic Committee of the School of Dentistry, Aichi-Gakuin University (approval number: 644). Prior to starting the study, a verbal explanation of the trial was provided, and written informed consent was obtained according to the guidelines of Aichi Gakuin University. Individuals without missing teeth were included in the study. Individuals with temporomandibular disorders or those taking antimicrobials were excluded from the study.

### Production and wearing protocol of experimental sports mouthguards
The impression of the dentition was obtained using a stock tray with alginate impression materials, and a stone model was fabricated. The EVA mouthguard sheet of 3 mm thickness was heated and pressed over the stone model using a vacuum pressure machine.

The participants wore a sports mouthguard for 2 h per day for 4 days. They were instructed to store it in a container under dry conditions after removing saliva using running water. They were also instructed to wear a sports mouthguard for at least 1 h after brushing their teeth after eating.

*Examination of cleaning effect on sports mouthguard*

The cleaning effect of the mouthguards was evaluated in two ways, *i.e.,* the degree of cleanliness using an inspection kit (Lucipak Pen; Kikkoman Bio Chemiphar, Tokyo, Japan) and bacterial count (CFU/ml). The colony count measures only the number of viable microorganisms, while the ATP + AMP wipe test measures the degree of contamination by measuring the amount of luminescence emitted by luciferase, a luminescent enzyme, and the amount of ATP+AMP converted, thereby measuring the presence of organic matter, including microorganisms.

The mouth guard was divided into two parts based on the median. Each of the two parts was treated with different processes: the right side was cleaned with only running water (CTRL) and the left side with a mouthguard cleaner (MC) for the *in vitro* experiments; cleaned at 40 °C for 10 min. Therefore, this study used a split-mouth design (*Pandis et al., 2013*).

The CTRL and MC groups were measured on the right premolar and contralateral sides, respectively, to assess the degree of cleanliness. The EVA surface was then swabbed with an inspection kit, and the degree of cleanliness was measured using an adenosine triphosphate (ATP)+ adenosine monophosphate (AMP) inspection device (Lumi Tester PD-30; Kikkoman Bio Chemiphar, Tokyo, Japan).

To assess the bacterial count, each specimen was washed with sterile distilled water, treated with an ultrasonic cleaner for 10 min in 30 ml of phosphate-buffered saline (PBS), and vortexed for 20 s to release the bacteria adhering to the specimens. The resulting bacterial suspension was inoculated and cultivated in BHI agar for total bacteria (under anaerobic conditions), Mitis-Salivarius agar for staphylococci (under anaerobic conditions), and Sabouraud agar for *Candida* (under aerobic conditions), and the number of bacterial colonies was counted. The reduction rate for each measured value was calculated using the following equation: ratios of MC vs. CTRL Lumi-Tester output = (value without MC cleaning -value with MC cleaning)/value without cleaning ×100 (%).

*Data analysis and statistics*

Data are expressed as mean ± standard deviation (SD). One-way analysis of variance (ANOVA) and Tukey's multiple comparison tests were performed using IBM SPSS statistics version 26.0 and results were considered statistically significant when $P$ values were <0.05.

# RESULTS

## Cleaning effect of MC on oral biofilm bacteria on EVA discs *in vitro*

We first examined the cleaning effect in the presence of biofilm bacteria, such as *S. oralis*, *S. mutans*, and *S. aureus*, on EVA discs. MC treatment significantly decreased *S. oralis*, *S. mutans*, and *S. aureus* counts compared to CTRL (Fig. 1). The bacterial counts (CFU/well) were below the detection limit after NaClO treatment. When treated with MC, the bacterial counts of *S. oralis* were reduced to the detection limit (bacteria were not detected).

The cleaning effects of oral bacterial biofilms on EVA discs were also monitored by SEM (Fig. 2). Biofilm formation on the discs of CTRL was intense for all the tested

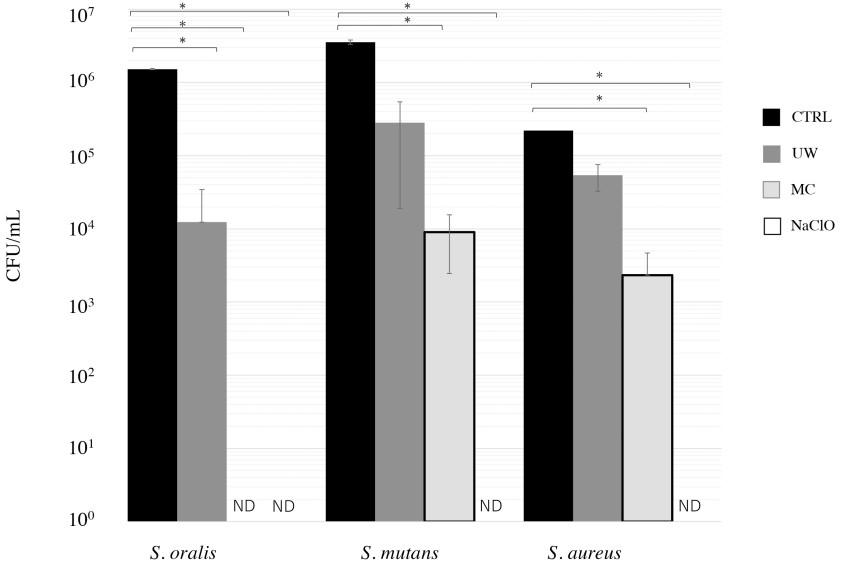

**Figure 1** **Cleaning effect of mouthguard cleaner (MC) on biofilm bacteria on ethylene-vinyl acetate (EVA) sheets.** Biofilms of *Streptococcus oralis* ATCC 9811, *Streptococcus mutans* NCTC 10449, and *Staphylococcus aureus* FDA209Pon EVA discs were treated with water (CTRL, black bars), ultrasonic washing (UW, dark gray bars), MC (right gray bars), or 250 ppm sodium hypochlorite (NaClO, white bars). After treatment, the remaining bacteria were measured in terms of colony-forming units (CFU) counts ($n = 6$). The vertical axis indicates CFU/ml. Data show mean ±SD. ND indicates that bacteria were not detected (below 2 CFU/well). An asterisk (*) denotes statistically significant differences at $P < 0.05$. Representative data for at least three independent experiments are shown.

bacterial strains. *S. oralis* formed sparse biofilms among the bacterial strains, whereas *S. mutans* and *S. aureus* cells formed dense ones. Biofilm formation was observed even in the grooves on the EVA surface. Biofilm formation on EVA decreased upon treatments with UW, MC, and NaClO for all bacteria. *S. oralis* appeared to be eliminated on EVA discs under NaClO treatment, whereas *S. mutans* and *S. aureus* slightly remained and were only completely eliminated under NaClO treatment conditions. The degree of reduction of the biofilm-forming bacteria of the three treatments was in the order of NaClO, MC, and UW.

### Cleaning effect of MC on experimental mouthguards worn *in vivo*

Next, we examined the disinfection effects on the experimental mouthguards. Here, we used two methods: the degree of cleanliness that requires bacterial contamination estimation using an ATP+AMP inspection device and the colony counts (CFU/well) to quantify total bacteria, *Streptococci*, and *Candida*.

Figure 3 shows the degree of cleanliness with or without MC treatment. In the MC group for each individual, ATP+AMP values were significantly reduced in nine of the ten subjects compared with the CTRL group (Fig. 3A). Notably, the MC treatment did not show any effect on the mouthguard obtained from subject D. In terms of the reduction rate of ATP+AMP values, the MC treatment resulted in a decrease of 63.5–99.7% in subjects, except for subject D (Fig. 3B).

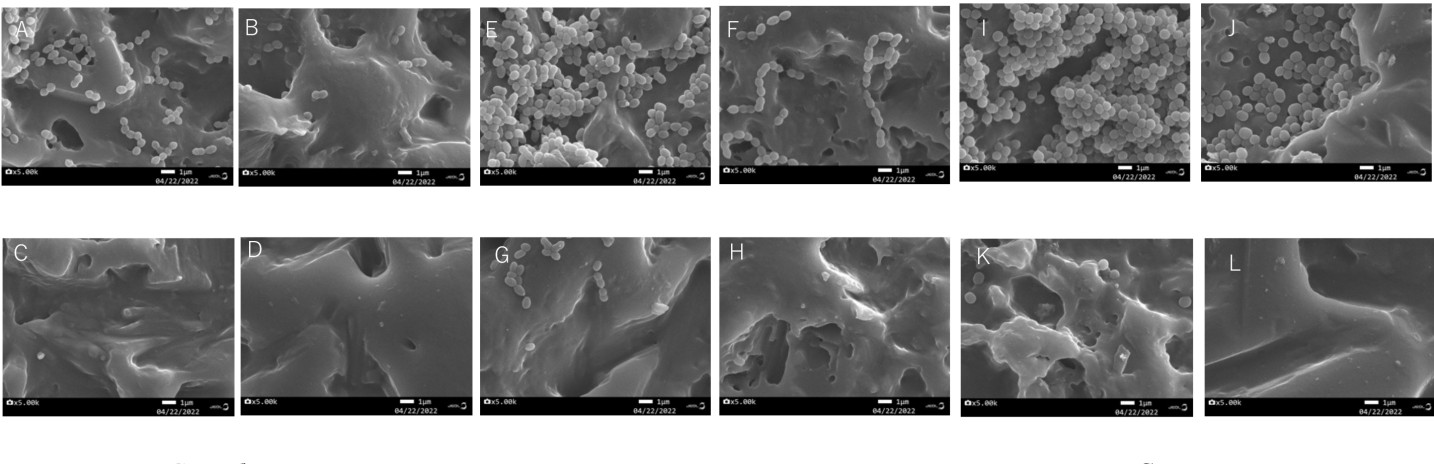

S. oralis          S. mutans          S. aureus

**Figure 2  Scanning electron microscope (SEM) images of bacterial biofilms formed on the surface of ethylene-vinyl acetate(EVA) sheets.**
Biofilms of *Streptococcus oralis* ATCC 9811, *Streptococcus mutans* NCTC 10449, and *Staphylococcus aureus* FDA209Pon EVA discs were treated with sterile water (CTRL), ultrasonic washing (UW) , mouthguard cleaner (MC), or 250 ppm sodium hypochlorite (NaClO), and observed by SEM. *S. oralis* (A: CTLR, B: UW, C: MC, D: NaClO); *S. mutans* (E: CTRL, F: UW, G: MC, H: NaClO); *S. aureus* (I: CTRL, J: UW, K: MC, L: NaClO). White bars indicate a width of 1 μm.

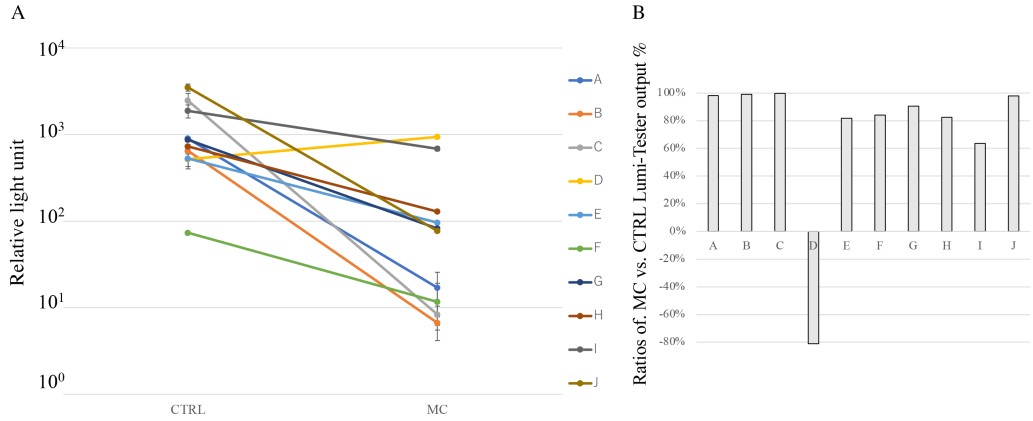

**Figure 3  Cleaning effect of mouthguard cleaner (MC) evaluated by the degree of cleanliness *in vivo*.**
(A) Cleanliness of oral bacteria for the used experimental sports mouthguards in individuals. Each line represents samples from the same individuals, which are shown in the same order as Fig. 4A (#A–J). The surface of experimental sports mouthguards with or without MC treatment was swabbed with an inspection kit, and the degree of cleanliness was measured with an ATP+AMP inspection device. The vertical axis indicates the relative light unit of cleanliness. There were significant differences between MC and CTRL treatments at $P < 0.05$ in all subjects. (B) Reduction rate of ATP+ADP value for each subject. The reduction rate for each measured value was calculated using the following equation: ratios of MC vs. CTRL Lumi-Tester output = (value without MC cleaning -value with MC cleaning)/value without MC cleaning × 100 (%).

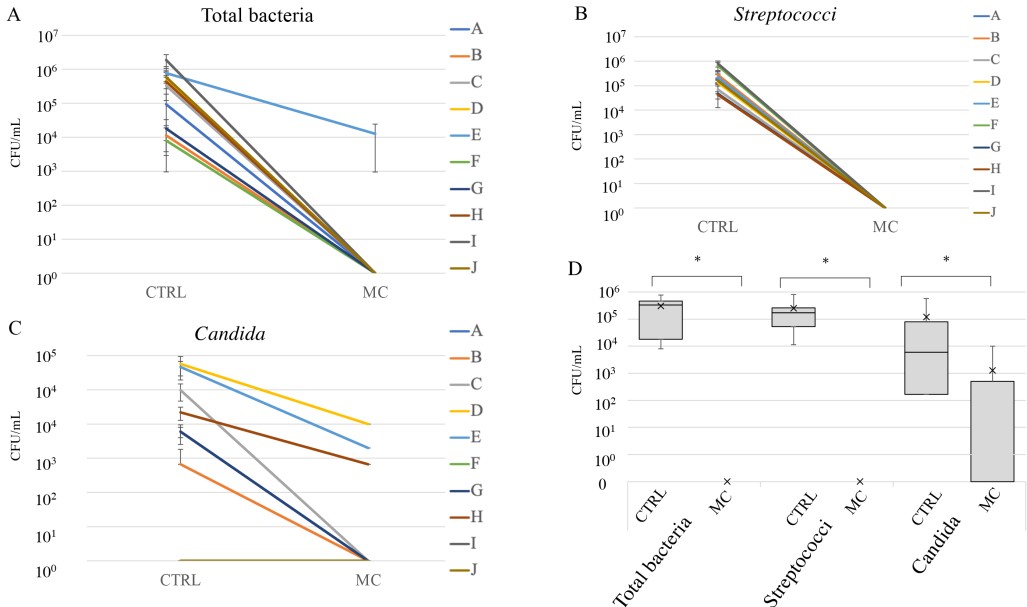

**Figure 4 Cleaning effect of mouthguard cleaner evaluated by bacterial count *in vivo*.** Counts of total bacteria (A), streptococci (B), and *Candida* (C) on the used experimental sports mouthguards in each individual. Each line represents samples from the same individuals, which are shown in the same order as in Fig. 3A (#A–J). The vertical axis indicates colony forming units (CFU)/ml. Box-and-whisker plots of overall microbial counts (D). The vertical axis indicates CFU/ml. An asterisk (*) denotes statistically significant differences at $P < 0.05$.

Figure 4 shows the microbial counts with or without MC treatment. Total bacterial and streptococcal counts of more than $1 \times 10^4$ CFU/ml were detected in all subjects without MC treatment, and *Candida* was detected in seven of the ten subjects. In all subjects, the counts of total bacteria, *Streptococci*, and *Candida* groups were significantly reduced with MC treatment (Figs. 4A–4C). However, for subjects D, E, and H, more than $10^3$ CFU/ml of *Candida* remained after MC treatments (Fig. 4C). In box-and-whisker plots of overall microbial counts, total bacteria, *Streptococci*, and *Candida* in the MC groups were significantly lower than those in the CTRL group (Fig. 4D).

## DISCUSSION

The cleaning effects of MC on oral bacterial biofilms formed on EVA discs were evaluated *in vitro* using CFU and SEM analyses. The results showed that the cleaning effect of MC was greater than that of UW (comparable to that of NaClO). In addition, MC treatment was sufficiently effective in cleaning oral microbiota, including streptococci and *Candida*, in experimental mouthguards designed to be used *in vivo*. These results collectively indicated that cleaning with running water or UW was insufficient for cleaning EVA mouthguards bound with oral bacteria and that cleaning with MC should be implemented.

NaClO has been accepted as a denture cleaning and disinfecting agent at various concentrations (*Felton et al., 2011*; *Fernandes, Orsi & Villabona, 2013*; *Rodríguez Acosta et al., 2015*; *Salloum, 2014*). It is also characterized by a broad antibacterial spectrum

(*Badaró et al., 2017*; *Rodríguez Acosta et al., 2015*; *Salles et al., 2015*). However, different concentrations and immersion periods may reduce the physical and mechanical properties of the materials and cause allergic reactions in susceptible individuals (*Chia Shi Zhe et al., 2016*; *Pisani et al., 2012*). Clinical studies are needed to evaluate whether a combination of lower concentrations and shorter immersion periods can reduce microbial biofilms without any disadvantages. Alternative agents also make for promising mouthguard cleansers, with good disinfecting action and negligible effects on biomaterials and allergic reactions.

Further in this study, two methods were used to evaluate the cleaning effect of MC on the used mouthguards: the colony count and the degree of cleanliness methods. Although there were some differences among the subjects based on both colony count and cleanliness result, the cleaning effects of MC was significantly higher than that of the CTRL treatment, corroborating the similarity of the results based on the colony count and the degree of cleanliness methods. The colony count requires time and effort for evaluation, while the degree of cleanliness using an inspection kit is a simple and rapid method (*Gillespie et al., 2017*; *Iwawaki et al., 2019*; *Watanabe et al., 2016*). Thus, the degree of cleanliness should be sufficient to evaluate the degree of mouthguard contamination in athletes busy with daily training.

In evaluating the colony counts in used mouthguards, $10^3$–$10^4$ CFU/ml of *Candida* remained even after the MC was treatment in subjects D, E, and H among the seven subjects in whom *Candida* was detected in the CTRL group (Fig. 4C). This high count of residual *Candida* may have affected the degree of cleanliness in these subjects (Figs. 3A and 3B). *Candida* is a commensal fungus that causes opportunistic infections and oral candidiasis in denture-wearing patients (*Fujinami et al., 2021*; *Morse et al., 2018*). Fungi are generally more resistant to various disinfectants than common bacteria (*Rutala, 1996*). If *Candida* is detected in the oral cavity of athletes, other additional treatments, such as strong disinfectants, may need to be selected. Our future plans include investigating the fungicidal effects of various disinfectants, including MC, on *Candida* spp.

Although the findings detailed above do seem robust, the number of subjects in our study was small. Further consideration of athletes commonly wearing sports mouthguards could increase the number of subjects in the future.

When sucrose was added to form strong biofilms on the EVA disc, more than $10^5$ and $10^6$ CFU/ml of *S. mutans* and *S. aureus*, respectively, were detected in CTRL. Although there were significant differences in the colony counts among the CTRL, MC, and UW groups, the UW and MC treatments left approximately $10^4$–$10^5$ and $10^3$–$10^4$ CFU/ml, respectively, suggesting the insufficiency of this treatment to clean the EVA disc. However, the total bacterial and streptococcal counts in the untreated experimental mouthguards were approximately $10^6$ CFU/ml, while those with the MC treatment were below the detection limit. These results suggest that MC treatment is sufficient for mouthguards worn for a short period, as demonstrated in this study. Lack of denture maintenance for long periods would lead to high contamination (*Coimbra et al., 2021*; *Tani et al., 2020*). In addition, *Ribeiro et al. (2021)* evaluated contamination of sports mouthguards worn three times a week for 1 h per day for 15 days. They reported that the sports mouthguards were contaminated after use, and a spray of chlorhexidine gluconate (0.12%) effectively

reduced contamination. As for orthodontic splints, which are similar devices as sports mouthguards, it has been revealed that many types of over-the-counter disinfectants are effective against oral bacteria, including *S. mutans*, *S. aureus* and *Candida*, that adhere to the devices (*Kiatwarawut, Rokaya & Sirisoontorn, 2022*). In recent years, graphene has attracted attention as a coating agent for dental devices with antimicrobial properties (*Srimaneepong et al., 2022*). These possible treatments will be the subject of further studies.

## CONCLUSIONS

Cleaning with MC was more effective than cleaning with UW. Enhanced cleaning may be a simple and economical method to promote oral hygiene in athletes. Since the study period was relatively short, future experiments should be conducted to simulate actual use in athletes that wear mouthguards for longer periods.

## ACKNOWLEDGEMENTS

We wish to thank Mariko Kondo for the support in the experiments. We wish to thank volunteers for providing the mouthguard samples. We wish to thank Satsuki Nagase for their support with the SEM observations. We wish to thank Editage for English language editing.

### Funding
This study was supported by JSPS KAKENHI grant No. 20K09931 (Yoshiaki Hasegawa). There was no additional external funding received for this study. The funders had no role in study design, data collection and analysis, decision to publish, or preparation of the manuscript.

### Grant Disclosures
The following grant information was disclosed by the authors:
JSPS KAKENHI: 20K09931.

### Competing Interests
The authors declare there are no competing interests.

### Author Contributions
- Hiroki Hayashi conceived and designed the experiments, performed the experiments, analyzed the data, prepared figures and/or tables, authored or reviewed drafts of the article, and approved the final draft.
- Yoshikazu Naiki conceived and designed the experiments, performed the experiments, analyzed the data, prepared figures and/or tables, authored or reviewed drafts of the article, and approved the final draft.
- Masahiro Murakami performed the experiments, analyzed the data, authored or reviewed drafts of the article, and approved the final draft.

- Akihiro Oishi performed the experiments, authored or reviewed drafts of the article, and approved the final draft.
- Rihoko Takeuchi performed the experiments, authored or reviewed drafts of the article, and approved the final draft.
- Masayoshi Nakagawa analyzed the data, authored or reviewed drafts of the article, and approved the final draft.
- Suguru Kimoto analyzed the data, prepared figures and/or tables, authored or reviewed drafts of the article, and approved the final draft.
- Yoshiaki Hasegawa conceived and designed the experiments, performed the experiments, analyzed the data, prepared figures and/or tables, authored or reviewed drafts of the article, and approved the final draft.
- Akizumi Araki conceived and designed the experiments, analyzed the data, authored or reviewed drafts of the article, and approved the final draft.

## Data Availability

The raw data are available in the Supplementary Files.

## Supplemental Information

Supplemental information for this article can be found online at http://dx.doi.org/10.7717/peerj.14480#supplemental-information.

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
