# Peer review of "Effects of cleaning sports mouthguards with ethylene-vinyl acetate on oral bacteria"

_PeerJ, doi:10.7717/peerj.14480_

## Round 0.1 · original submission · Minor Revisions

Befrore sending your manuscript for review, I must ask you to perform a few changes to improve readability:

1) there is some confusion in the language, where at times you write "cleanliness" as a synonym of "number of CFU", when it is actually the opposite. For example: in the abstract, you state "The degree of cleanliness of the used experimental sports mouthguard in MC was significantly lower than that in CTRL." which means that in MC the mouthguard remained dirtier. Your data show that the mouthguard becomes cleaner in MC than CTRL, though. Please correct this throughout (e.g. line 216)

2) Graph 3.B is quite puzzling, as it shows three points where there is an increase in Lumi-Tester output, whereas in panel A there was only one subject where Lumi-Tester output increased upon MC treatment.Please redraw panel B with ratios of MC vs. CTRL Lumi-Tester output (rather than reduction rates), instead.

3) Figure 4A is unfortunately quite unreadable, since it is very hard to tell apart the shadings of the bars even after magnification. Please use an appropriate color code, increase the size of the graph to full page width or use some other means to make its reading comfortable. In Figure 4B, there are (at present) no labels to tell apart CTRL from MC (although I admit it is obvious to interpret, for someone what has been only mildly attentive to the paper). Please add such labels. I suggest 4B might be more informative with a logarithmic scale.

---

## Round 0.2 · Minor Revisions

Please address the issues highlighted by our thoughtful reviewers.

·

Basic reporting

In this research, the researchers evaluated the cleaning effects of eûciency with a mouthguard cleaner (MC) on microbial bioûlm formation in sports mouthguards in vitro and in vivo in 3 bacteria: Streptococcus oralis, Streptococcus mutans, and Staphylococcus aureus. And they found that the number of bacterial cells significantly decreased against all the tested biofilm bacteria upon treatment with mouthguard cleaner, compared with sterile distilled water and ultrasonic washing.

The title is confusing. Can the title be better edited as Cleaning Effects of Ethylene-Vinyl Acetate Sports Mouthguard on Oral Bacteria?

Are the bacteria (Streptococcus oralis, Streptococcus mutans, and Staphylococcus aureus) studied in this study present normally? Please mention.

In the abstract, the method is not too clear. In the materials and method, the authors need to describe invivo study and invitro study.
Is the mouthguard cleaner made from ethylene-vinyl acetate (EVA)? Need to mention it clearly.
In conclusion, it is better to add the content of MC.

Experimental design

How the sample size calculation is done? It is better to add.

The subheadings need to be improved.

For the analysis, which software is used?

Validity of the findings

Values of MC are not visible in Figure 1. So, it is better to add the Table of values in Figure 1.
Increase the magnification of the SEM Figures for better visualization.

Additional comments

Mouth guards are similar to the ortho splint. Please discuss.
https://pubmed.ncbi.nlm.nih.gov/35683929/

It is better to add the potential newer antimicrobial agents for the mouthguard.
e.g. graphene coated prostheses
https://pubmed.ncbi.nlm.nih.gov/35008923/

Please add the limitations of this research.

·

Basic reporting

I think the data could be better presented to make it more easily and quickly understood by the audience:

Figure 3A. I think this figure would be much more effective if it was a stacked line graph rather than a very busy bar chart (see attached). Given they are all significantly different, you don’t need to add a star for each, you can just cite this in the legend.

Figure 3B. I don’t understand this graph. Shouldn’t it be the ratio of the readings for each subject rather than all the subjects combined?

Figure 4A. I think this figure would be much more effective if the data was split into 3 graphs – one for total bacteria, one for candida and one for streptococci, and each one presented as stacked line graphs comparing CTRL and MC as suggested for Fig. 3A.

Figure 4B. Please convert this to a log scale to better show the data.

Minor corrections:

Line 225: The authors write “The reduction rates of total bacteria, Streptococci, and Candida in MC groups were significantly lower than those in the CTRL group (Fig. 4B)”. I think this sentence should be reworded as its not clear if this means CTRL was better or worse than MC. If the reduction rates were lower this could be interpreted as the reduction was lower in MC compared to CTRL which is the opposite to what the data shows.

Figure 1: the legend has MC as white bars and sodium hypochlorite as grey bars but these are reversed in the legend provided on the graph. Which is correct?

Experimental design

The experimental design needs some clarification to explain biological and technical replicates:

Line 113. Can you clarify how you did the experiments and provide more information on biological and technical replicates? At the moment it could be interpreted as the discs were inoculated with a mix of the three bacteria and then divided between the four treatments. This would give 6 technical replicates with the experiment performed once (1 biological replicate). Or were there 24 discs for each bacterium, divided between the four treatments? If so, was this just done with one biological replicate for each bacterium?

Line 132. What was the rationale for incubating the bacteria for 24h rather than enumerating them immediately?

Validity of the findings

Fine.

Reviewer 3 ·

Basic reporting

First of all, thank you for considering me for reviewing the article entitled" Cleaning Effects of Ethylene-Vinyl Acetate on Oral Bacteria in Sports Mouthguard ". The objective of this manuscript is interesting. Although the first reviewer’s comments are mostly addressed, there are still some minor concerns in terms of the language, referencing style, and writing style based on the comments below. I hope my comments are helpful:
(The lines numbers are based on the pdf document)

Introduction:

- lines 73,74 & 83,84: Please move all the knowledge gaps together before the last paragraph of the introduction. Move the sentence “ However, solid evidence on the efficiency of cleaning methods for sports mouthguards is lacking.” Before this sentence, “However, no cleaning method with standardized guidelines has been developed thus far in Japan.” Please ensure that you keep the flow of the content and paragraphs while doing this.

- Line 81 : It is unnecessary to repeat the in-text citation when you are already mentioning the author's name and the publication year. But make sure that the reference is included in the reference list. For example: “ Glass et al., 2011 reported that daily cleaning with cleaning agents could reduce microbial contamination on sports mouthguards (Glass et al., 2011). “

Materials & Methods:

- Line 170: Please change “ cleaned was performed at 40°C for 10 min” to “cleaned at 40°C for 10 min”.

- Line 178: Please define “ PBS” before using its abbreviation in the text

Experimental design

no comment.

Validity of the findings

no comment.

---

## Round 0.3 · Minor Revisions

Please include the critical information requested by reviewer #2. Regarding the title, I think the current version (suggested by one of the reviewers) is not grammatical, and I would prefer the original title.

·

Basic reporting

The overall manuscript has been improved including the title.

Experimental design

no comments

Validity of the findings

The findings are explained well.

Additional comments

It can be accepted in the present form.

·

Basic reporting

No comment

Experimental design

The manuscript is still lacking important details about the in vitro experiments. In my previous review, I asked the authors to clarify how they did the in vitro experiments and provide more information on biological and technical replicates. This is what I wrote: At the moment it could be interpreted as the discs were inoculated with a mix of the three bacteria and then divided between the four treatments. This would give 6 technical replicates with the experiment performed once (1 biological replicate). Or were there 24 discs for each bacterium, divided between the four treatments? If so, was this just done with one biological replicate for each bacterium?

In their rebuttal, the authors wrote: "We added information on biological and technical replicates in lines 120 and 126-127."

This information hasn't been added. At line 20 they have just added the words "For S. mutans" and at lines 126-127 they have added "For S oralis and S. aureus, bacterial biofilms were formed byin the same procedure as for S. mutans."

The authors need to specify how many biological and technical replicates they used for each microbe.

Validity of the findings

No comment

Reviewer 3 ·

Basic reporting

Thank you again for considering me for reviewing the article entitled" Cleaning Effects of Ethylene-Vinyl Acetate on Oral Bacteria in Sports Mouthguard ".

All my previous comments are addressed and I do not have any more comments.

Experimental design

no comments.

Validity of the findings

no comments.

---

## Round 0.4 · accepted · Accept

Thank you for addressing the final issues.